# Photodynamic Therapy with Nebulized Nanocurcumin on A549 Cells, Model Vessels, Macrophages and Beyond

**DOI:** 10.3390/pharmaceutics14122637

**Published:** 2022-11-29

**Authors:** María Julia Altube, Ezequiel Nicolás Caputo, Martín Nicolás Rivero, María Laura Gutiérrez, Eder Lilia Romero

**Affiliations:** 1Nanomedicine Research and Development Centre (NARD), Science and Technology Department, National University of Quilmes (UNQ), Roque Saenz Peña 352, Bernal B1876, Argentina; 2Institute of Pharmacology, University of Buenos Aires (UBA), Consejo Nacional de Investigaciones Científicas y Técnicas (CONICET), Paraguay 2155 piso 9, Buenos Aires C1121, Argentina

**Keywords:** curcumin, PDT, nanoarchaeosomes, A549, macrophages, vessels

## Abstract

This study aimed to determine the damage mechanisms caused by naturally targeted nanoarchaeosomes made of diether lipids from *Halorubrum tebenquichense* loaded with curcumin (CUR, nATC), which mediated photodynamic therapy (PDT) on A549 cells and on THP-1-macrophages, two cell types found in airway cancers. The effect of nATC- PDT on vessels modeled with a chicken embryo chorioallantoic membrane (CAM), after dropping the formulations on its surface covered with mucins, was also determined. nATCs are known to efficiently trap CUR for at least six months, constituting easy-to-prepare, stable formulations suitable for nebulization. CUR instead, is easily released from carriers such as liposomes made of ordinary phospholipids and cholesterol after a few weeks. Irradiated at 9 J/cm^2^, nATC (made of archaeolipids: Tween 80: CUR at 1:0.4:0.04 *w*:*w*, size 180 ± 40 nm, ζ potential −24 mV, 150 μg CUR/15 mg lipids/mL) was phototoxic (3.7 ± 0.5 μM IC_50_), on A549 cells after 24 h. The irradiation reduced mitochondrial membrane potential (ΔΨm), ATP levels and lysosomal functionalism, and caused early apoptotic death and late necrosis of A549 cells upon 24 h. nATC induced higher extra and intracellular reactive oxygen species (ROS) than free CUR. nATC-PDT impaired the migration of A549 cells in a wound healing assay, reduced the expression of CD204 in THP-1 macrophages, and induced the highest levels of IL-6 and IL-8, suggesting a switch of macrophage phenotype from pro-tumoral M2 to antitumoral M1. Moreover, nATC reduced the matrix metalloproteinases (MMP), −2 and −9 secretion, by A549 cells with independence of irradiation. Finally, remarkably, upon irradiation at 9 J/cm^2^ on the superficial vasculature of a CAM covered with mucins, nATC caused the vessels to collapse after 8 h, with no harm on non-irradiated zones. Overall, these results suggest that nebulized nATC blue light-mediated PDT may be selectively deleterious on superficial tumors submerged under a thick mucin layer.

## 1. Introduction

CUR is a hydrophobic polyphenol without nutraceutical value, and is used as a food additive [1]. According to the US Food and Drug Administration (FDA), 95% *w/w* curcuminoids (comprising CUR, bisdemethoxycurcumin, and demethoxycurcumin) at doses up to 12 g/day have the status of “Generally Recognized as Safe” (GRAS) [2]. CUR, on the other hand, has been known from ancient times to act against a variety of pathological conditions; its use against cancer, inflammatory, neurological, cardiovascular, and skin diseases is well documented in current academic settings. The exploitation of such therapeutic potential is hampered, however, by CUR’s structural lability to environmental conditions (photodegradation), even in aqueous media at physiological pH [3], its poor oral absorption [4], and its fast degradation in blood (administered by oral route, its inert conjugates are rapidly excreted, and systemically is transformed into hydro derivatives while some evidence indicates its degradation leads to products with useful biological activity [5]. Today, there is a uniform consensus about the inclusion of CUR into nanoparticles of variable chemical, size, and surface nature, as an efficient strategy to protect CUR’s structure, increase its bioavailability, and modulate its pharmacokinetics [6]. In other words, CUR-based nanomedicines may enable, in the future, the pharmaceutical use of CUR.

Photosensitization is part of the pleiotropic activity of curcuminoids [7,8,9]. The photodynamic mechanism utilizes a non-toxic chromophore, known as a photosensitizer (PS), which is excited from its ground singlet state (S_0_) to an excited singlet state (S_n_) with appropriated wavelength (typically from the visible to near infrared spectrum). Afterwards, the PS can lose energy by emitting fluorescence or heat via internal conversion, thus returning to S_0_, or it can be transformed to a longer living excited triplet state PS (T_1_) by so-called inter-system crossing. From this state, it can return to its S_0_ either by phosphorescence emission or by two mechanisms generating ROS, as follows: in type I mechanisms, a charge (i.e., electrons) is transferred to surrounding substrates leading to formation of superoxide radicals (O2^•−^) that can undergo dismutation into hydrogen peroxide (H_2_O_2_), the precursor of the highly reactive free hydroxyl radicals (HO^•−^), which are formed via Fenton-like reactions. In contrast, in type II mechanisms, energy (but no charge) is transferred directly to the ground state molecular oxygen (^3^O_2_), originating energized molecular oxygen, known as singlet oxygen (1O_2_) [7,10,11].

However, CUR is not excitable with light from the first therapeutic window at 650–850 nm, capable of penetrating several centimeters within biological tissues [NIR radiation] [12], since its absorption spectra display an intense peak in the blue zone at 420 nm [13,14], coincident with the Soret band at 425 nm from blood porphyrins [15]. CUR- mediated PDT, therefore, is of limited therapeutic application: ~1% of 450 nm light reaches a depth of around 1.6 mm; 1% of wavelength higher than 650 nm light instead, penetrates about 4.75–5 mm [16,17]. Because of these reasons, CUR-mediated PDT is restricted to the treatment of superficial targets, such as skin infections [18]; and was recently proposed as food preservation technology [19]. Even superficial tumors on the skin display a three-dimensional morphology and their volumes penetrate several microns and up to millimeters below the epithelia [20]. In such settings, the fact that blue light is absorbed by the microvascular bed from subcutaneous tissue in the skin and does not efficiently penetrate beyond the stratum corneum reduces the efficiency of PDT to treat skin tumors. CUR-mediated PDT, however, may be used to treat epithelial tumors where light absorption across the skin is not a limiting factor [21]. Lung epithelial tumors lack stratum corneum, and the tumor cells are relatively exposed to the light of the respiratory tract. PDT is an approved therapy for the ablation of endobronchial tumors [22]. In such treatments, the photosensitizer is intravenously administered to directly contact the intima (one main effect of PDT is the damage of the vasculature that feeds the tumor) and then diffuse or extravasate into the tumor microenvironment. The design of nebulized formulations of CUR for antitumoral PDT has recently been addressed by Bakowski [23]. Nebulized formulations of CUR should be able to act on tumor cells, their microenvironment, and be available for vasculature when delivered from the outer side of the vessels.

Our group has previously reported the performance of nATC, nebulized nanovesicles solubilizing ~600 µM CUR, prepared with polar lipids from the archaea *H. tebenquichense*, naturally targeted to cells expressing the SRA1. The loading of considerable amounts of CUR (150 μg CUR/15 mg lipids/mL) required the addition of Tween 80 to the lipid bilayers. By SAXS and Raman, it was determined that CUR, surrounded by Tween 80 molecules, experienced a tight immobilization interacting with the terminal side of diether archaeolipid chains of high microviscosity and low lateral mobility. Such interaction was absent in liposomes, which slowly released CUR along storage. nATC resulted not only in being structurally stable in front of nebulization stress, but importantly, also against dilution and storage for months. Nebulized at 5 μg CUR/mL on an inflamed air–liquid interface made of epithelial A549 cells (a cell line used as non-small lung tumor cells (NSLTC) model), nATC increased TEER, normalized the permeation of Lucifer Yellow, and decreased the release of IL-6, TNF-α, and IL-8 [24]. In other words, nATC magnified the anti-inflammatory activity of CUR and repaired the epithelial damage (this last effect is exclusively owed to the archaeolipid matrix) without inducing cytotoxicity. The effect of PDT on photosensitizers delivered on the surface of vessels and of tumor microenvironment, is currently unknown. To the best of our knowledge, this is the first report exploring the PDT of CUR in archaeosomes made of diether lipids, upon being endocytosed by A549 cells, by THP-1 monocytes derived macrophages and on a superficial vasculature, modeled with a chick embryo CAM.

## 2. Materials and Methods

### 2.1. Materials

CUR, cholesterol (chol), 3-(4,5-dimetiltiazol-2-yl)-2,5-diphenyl tetrazolium bromide (MTT), phorbol 12-13-acetate (PMA), 2-mercaptoethanol, trizma base, Tween 80, Tween 20, Triton X-100, sodium dodecyl sulfate (SDS), and ammonium persulfate were from Sigma-Aldrich (St. Louis, MO, USA). 1,2-hydrogenated-L-α-phosphatidylcholine (HSPC) was from Northern Lipids Inc. (Vancouver, BC, Canada). Roswell Park Memorial Institute medium (RPMI), penicillin–streptomycin sulphate, glutamine, sodium pyruvate, and trypsin/ethylenediamine tetraacetic acid were from Thermo Fisher Scientific (Waltham, MA, USA). Acrylamide, bis-acrylamide, and TEMED were obtained from Bio-Rad Laboratories (Hercules, CA, USA). Fetal bovine serum (FBS) was from Internegocios (Buenos Aires, Argentina). Yeast extract, Griess Reagent Solution “A” and Griess Reagent Solution “B” were from Laboratorios Britania S.A. (Buenos Aires, Argentina). The other reagents were of analytic grade from Anedra, Research AG (Buenos Aires, Argentina).

### 2.2. Archaebacteria Growth, Extraction, and Characterization of Total Polar Archaeolipids (TPA)

*H. tebenquichense* archaea, isolated from soil samples of Salina Chica, Peninsula de Valdés, Chubut, Argentina were grown in basal medium supplemented with yeast extract and glucose [25] in a 25-L homemade stainless-steel bioreactor at 40 °C and harvested 72 h after growth. TPA were extracted from biomass using the Bligh and Dyer method modified for extreme halophiles [26]. Around 700 mg TPA were isolated from each culture batch. The reproducibility of each TPA extract composition was routinely screened by phosphate content [27] and electrospray–ionization mass spectrometry [28].

### 2.3. Preparation of CUR Nanovesicles

Conventional nanovesicles containing Tween 80 (HSPC: chol: Tween 80 1:0.33:0.53 *w*:*w*, nLT), nLT loaded with CUR (HSPC: chol: Tween 80: CUR at 1:0.33:0.53:0.053 *w*:*w*, nLTC), TPA-nanovesicles containing Tween 80 (TPA: Tween 80, 1:0.4 *w*:*w*, nAT) and nAT loaded with CUR (TPA: Tween 80:CUR at 1:0.4:0.04 *w*:*w*, nATC) were prepared by the film hydration method, and were homogenized by sonication and extrusion as described before [24]. The resulting nanovesicles were sterilized by passage through a 0.22 µm sterile filter and stored at 4 °C protected from light.

### 2.4. Structural Characterization of CUR Nanovesicles

Phospholipids were quantified by a colorimetric phosphate microassay [27]. CUR was quantified by absorbance at 425 nm upon complete disruption of one volume of nanovesicles suspension in 150 volumes of methanol [24]. Size and ζ potential were determined by dynamic light scattering and phase analysis light scattering, respectively, using a nanoZsizer apparatus (Malvern Instruments, Malvern, UK).

### 2.5. LED Irradiation Device

A homemade LED irradiation device with a wavelength of 420 nm was used at a power of 30 mW/cm^2^ (Appendix A). The cell culture plates or samples were fit into the device at a distance of 10 cm from the LED. Radiation fluence was calculated as a function of radiation time.

### 2.6. Cell Lines and Culture Conditions

Human epithelial lung cell line A549 (ATCC^®^ CCL-185™) was maintained in RPMI supplemented with 10 % FBS, 100 U/mL penicillin, 100 µg/mL streptomycin, and 2 mM L-glutamine. The human monocyte cell line THP-1 (ATCC TIB-202™) was maintained in RPMI supplemented with 100 U/mL penicillin, 100 µg/mL streptomycin, 2 mM L-glutamine, 0.05 mM 2-mercaptoethanol and 1 mM sodium pyruvate. THP-1 monocytes were differentiated into macrophages by treatment with 100 ng/mL PMA for 24 h. All cell lines were grown in a humidified atmosphere of 5% CO_2_ at 37 °C.

### 2.7. Fluorescence, Absorbance Spectra, and Induction of ROS and Reactive Nitrogen Species (RNS)

Fluorescence emission measurements λex = 425 nm and absorption measurements of free CUR and CUR nanovesicles at 40 μM in RPMI supplemented with 5% SFB before and after samples irradiation (fluence of 9 J/cm^2^) were obtained with Cytation™ 5 (BioTek Instruments, Inc., Winooski, VT, USA).

For measuring extracellular ROS under cell-free conditions, 1 mM of the general oxidative stress indicator chloromethyl-2′,7′-dichlorodihydrofluorescein diacetate (CM-H2DCFDA), (Thermo Fisher Scientific, Waltham, MA, USA) was deacetylated with 0.01 M NaOH at 37 °C for 30 min protected from light [29]. Free CUR or CUR nanovesicles were diluted to 20 and 40 μM in a 5 μM solution of CM-H2DCF in Tris buffer pH 7.4. After 30 min incubation fluorescence intensity was measured at λex = 490 nm and λem = 530 nm in a Cytation™ 5. A positive control of ROS production was carried out by incubating 400 μM of hydrogen peroxide with the fluorescent probe.

ROS or RNS generation by A549 cells after PDT was measured using CM-H2DCFDA dye and Griess method, respectively. Briefly, A549 cells were seeded in 48-well plates with a density of 2 × 10^4^ cells per well and grown for 24 h. Then, cells were incubated with nATC, nLTC or free CUR at concentrations of 20 and 40 μM in RPMI supplemented with 5% FBS. Cells were incubated at 37 °C to allow nanovesicles uptake, then plates were irradiated with a fluence of 9 J/cm^2^ or were incubated in the dark (non-irradiated). After 24 h incubation at 37 °C in a 5% CO_2_ atmosphere, supernatants were collected for RNS quantification and cells were washed with PBS. ROS was quantified in culture cells by adding 4.4 μg/mL CM-H2DCFDA dye in PBS. After 30 min incubation at 37 °C in a 5% CO_2_ atmosphere, cells were washed. Fluorescence intensity at λex = 492 nm and λem = 517 nm was determined in a Cytation™ 5. A positive control of ROS production was carried out by incubating cells with 400 μM of hydrogen peroxide. RNS was quantified in culture supernatants after precipitating cells and debris, by adding 1 volume of Griess Reactive Solution “A” and 1 volume of Griess Reactive Solution “B”. After 15 min incubation protected from light, the absorbance was measured at 525 nm in a Cytation™ 5 instrument. RNS concentrations were calculated by comparison with absorbance at 525 nm of standard calibration solutions made with sodium nitrite from 2.5 to 30 μM.

### 2.8. Cytotoxicity of CUR Nanovesicles upon Irradiation

Cell viability upon 24 h of incubation with nanovesicles was measured by the MTT assay. Briefly, A549 cells were seeded in 96-well plates with a density of 1 × 10^4^ cells per well and grown for 24 h. Then, cells were incubated with CUR nanovesicles or free CUR, (prepared from a stock of 13,6 μM CUR in dimethyl sulfoxide (DMSO)), at concentrations of 0.625, 1.25, 2.5, 5, 10, 20, and 40 μM of CUR in RPMI supplemented with 5% FBS. DMSO at 0.3% *v*/*v* and empty nanovesicles at total lipid concentrations of 0.25, 0.5, 1, and 3 mg/mL were also tested. Cells were incubated at 37 °C for 1 h to allow nanovesicle cellular uptake and then plates were irradiated for 1, 5, or 10 min, corresponding to a fluence of 1.8, 9, and 18 J/cm^2^, respectively. For the non-irradiated control, plates were incubated without irradiation. For washing the control, after 1 h incubation with samples, cell supernatants were removed, washed twice with PBS and plates were irradiated with a fluence of 9 J/cm^2^. After irradiation, cells were incubated for 24 h at 37 °C in a 5% CO_2_ atmosphere, then the medium was removed, cells were washed with PBS and 100 µL of 5 mg/mL MTT solution were added to each well. After 2 h incubation at 37 °C, MTT solution was removed, the insoluble formazan crystals were dissolved in DMSO, and absorbance was measured at 570 nm using Cytation™ 5. The cell viability was expressed as a percentage of cells untreated and IC_50_ values were determined for each condition applying a non-linear regression model of dose-response curve, (Inhibitor) vs. normalized response with GraphPad Prism 8.0.1 (Graph Pad Software, Inc., San Diego, CA, USA).

### 2.9. ΔΨm

The effect of CUR nanovesicles on ΔΨm of A549 was determined using JC-1 dye mitochondria staining kit (Sigma-Aldrich, St. Louis, MO, USA) according to the manufacturer guidelines. Briefly, A549 cells were seeded in 48-well plates with a density of 2 × 10^4^ cells per well and grown for 24 h. Then, cells were incubated with nATC, nLTC or free CUR at concentrations of 5, 10, 20, and 40 μM in RPMI supplemented with 5% FBS. Cells were incubated at 37 °C to allow nanovesicles uptake, then plates were irradiated with a fluence of 9 J/cm^2^ or incubated in the dark (non-irradiated). After 24 h incubation, supernatants were discarded, cells were washed with PBS and incubated with 2.5 μg/mL JC-1 staining mixture for 15 min at 37 °C. Upon supernatant removal and PBS washing, the fluorescence intensity of each well was measured in Cytation™ 5. The fluorescence of JC-1 monomers was determined at λex = 490 nm and λem = 530 nm and JC-1 aggregates at λex = 525 nm and λem = 590 nm. The positive control was carried out by incubating cells with a solution of 100 ng/mL of valinomycin.

### 2.10. Apoptosis/Necrosis Assays

Induction of apoptosis/necrosis on A549 cells by PDT was determined by staining with Annexin V, propidium iodide (PI) or YO-PRO™-1 dyes (Thermo Fisher Scientific, Waltham, MA, USA). A549 cells were seeded at a density of 3.2 × 10^5^ cells per well in 12-well plates and grown for 24 h. Then, cells were incubated with CUR nanovesicles or free CUR at concentrations of 20 and 40 μM in RPMI supplemented with 5% FBS for 1 h at 37 °C and cells were irradiated with a fluence of 9 J/cm^2^ and incubated for 1, 6 or 24 h. For Annexin V/PI staining, after 1 and 6 h incubation, cells were trypsinized, washed with PBS, and resuspended in 20 μL of binding buffer. Then, cells were incubated with fluorescein-conjugated Annexin V reagent for 15 min at room temperature in the dark. Thereafter, cells were washed and resuspended on binding buffer with PI for 15 min at 4 °C. A total of 1 × 10^4^ cells were analyzed using a FACSCalibur flow cytometer (Becton Dickinson, San José, CA, USA) and data analysis was performed with the FlowJo 10.8.0 software (Flowjo, LLC, Ashland, OR, USA). Annexin V-positive cells were considered apoptotic, whereas Annexin V and PI-positive cells were considered necrotic. For YO-PRO™-1 and PI stanning, after 24 h incubation, cell supernatants were discarded, washed, and incubated at 0.5 μM YO-PRO™-1 and incubated for 15 min at room temperature. After that, PI was added to each well and incubated for 15 min at 4 °C and washed with PBS. Fluorescence microscopies were made with Cytation™ 5. YO-PRO™-1 positive cells were considered apoptotic, whereas YO-PRO™-1 and PI-positive cells were considered necrotic.

### 2.11. Intracellular ATP Content

The intracellular ATP content of A549 cells after PDT was determined with CellTiter-Glo^®^ Luminescent Cell Viability Assay (Promega, Madison, WI, USA) according to manufacturer guidelines. Briefly, A549 cells were seeded and incubated with nATC, nLTC, or free CUR as stated in Section 2.9. After 24 h incubation at 37 °C in a 5% CO_2_ atmosphere, supernatants were discarded, and cells were washed with PBS and incubated with fresh medium for 30 min at room temperature. Then, one volume of CellTiter-Glo^®^ reactive was added to cell media in each well, stirred for 2 min in an orbital shaker, and incubated for 10 min at room temperature until signal stabilization. The luminescence of each well was measured in a Cytation™ 5.

### 2.12. Lactate Dehydrogenase (LDH) Leakage

The LDH release on A549 cells supernatant after PDT was determined with CytoTox 96^®^ Non-Radioactive Cytotoxicity Assay (Promega, Madison, WI, USA), according to manufacturer guidelines. Briefly, A549 cells were seeded and incubated with nATC, nLTC, or free CUR as stated in Section 2.8. Cells were incubated at 37 °C to allow nanovesicles uptake, then plates were irradiated with a fluence of 9 J/cm^2^ or incubated in the dark (non-irradiated). After 24 h incubation at 37 °C in a 5% CO_2_ atmosphere an aliquot of 35 μL supernatants was transferred to fresh 96-well plates, 35 μL of the CytoTox 96^®^ Reagent were added to each well and incubated for 30 min at room temperature. The reaction was terminated by adding 35 μL of Stop solution. The absorbance was measured at λ 490 nm in Cytation™ 5.

### 2.13. CUR Cellular Uptake and Lysosomal Damage

CUR cellular uptake was measured by fluorescence confocal microscopy. LysoTracker™ Red DND-99 (LR), (Thermo Fisher Scientific, Waltham, MA, USA), a fluorescent probe that accumulates in acidic compartments such as lysosomes, was used as an indicator of lysosomal health, as well as a marker to study CUR colocalization in lysosomes. A549 cells were seeded at a density of 1.2 × 10^5^ cells per well in 12-well plates with rounded cover slips on the bottom and allowed to attach overnight. Then, cells were incubated with CUR nanovesicles or free CUR at concentrations of 40 μM in RPMI supplemented with 5% FBS for 1 h or 6 h at 37 °C. For LR assays samples were incubated for 1 h, irradiated with a fluence of 9 J/cm^2^, and incubated for an additional 5 h. Then supernatants were discarded, cells were washed with PBS, and incubated with 3.8 µM LR for 30 min at 37 °C for endosomal–lysosomal staining. Afterward cells were washed and fixed for 5 min with formaldehyde 3.75% (*w/v*) in PBS. For cellular uptake determinations, samples were incubated for 1 h at 37 °C, washed with PBS and cells were fixed. Finally, preparations were washed three times in PBS and mounted in a 90% (*w/v*) glycerol solution. Fluorescence microscopies were made with a Leica laser-scanning spectral confocal microscope TCS SP8; (Leica Microsystem, Wetzlar, Germany). Pearson’s correlation coefficient (Rr), which describes the correlation of intensity distribution between channels, was calculated using Fiji Software [30]. The degree of colocalization was determined from Rr values between −0.2–0.09 (weak colocalization), 0.1–0.48 (moderate colocalization), and 0.49–0.84 (strong colocalization) [31].

### 2.14. Tumor-Associated Macrophages (TAM) Modulation

TAM-like macrophages were generated by cultivating THP-1 macrophages with A549 cell culture supernatant [32]. THP-1 cells were seeded in 24-well plates with a density of 6 × 10^4^ cells per well in RPMI supplemented with 10% FBS, pyruvate sodium 1% (*w/v*), 100 ng/mL of PMA, and incubated for 24 h to allow macrophage transformation. A549 cells were seeded at a density of 2 × 10^5^ cells per well in 48-well plates and grown for 24 h. Then, A549 cells were incubated with CUR nanovesicles or free CUR at concentrations of 5 μM and 40 μM for 1 h at 37 °C and were irradiated with a fluence of 9 J/cm^2^ or left in the dark (non-irradiated), and incubated for additional 24 h. Then 250 μL of A549 supernatants were transferred to each THP-1 well and 50 μL of fresh RPMI medium was added. After 24 h, the supernatants were stored at −20 °C for cytokine determination. THP-1 cells were washed with PBS, trypsinized, washed again, and labeled with the CD204-APC antibody (Thermo Fisher Scientific, Waltham, MA, USA) for 30 min, except for the control cells. Cells were then washed with PBS and fixed with formaldehyde 3.75% (*w/v*) in PBS for 10 min at 4 °C. A total of 2 × 10^4^ cells were analyzed using a FACSCalibur flow cytometer (Becton Dickinson, San José, CA, USA), and data analysis was performed with FlowJo software. Human IL-8 and IL-6 concentrations were measured by enzyme-linked immunosorbent assay BD OptEIA™ (BD Biosciences, San Jose, CA, USA) following the manufacturer’s instructions. Absorbance measurements were carried out at 450 nm on a microplate reader in a microplate reader.

### 2.15. Migration Assay

A549 cells were seeded at a density of 3.2 × 10^5^ cells per well in 12-well plates and grown for 48 h to complete confluence in RPMI supplemented with 10% FBS. Scratch wounds were created in confluent monolayers using a sterile p200 pipette tip. Following washing, cells were cultured in RPMI without SFB and incubated with CUR nanovesicles or free CUR at concentrations of 5 μM. After that, cells were irradiated with a fluence of 9 J/cm^2^ and incubated at 37 °C. After 24 h incubation, cells were washed with PBS and incubated with RPMI without SFB for an additional 72 h. Repopulation of the wounded areas was observed under phase-contrast microscopy with Cytation™ 5. The size of the scratch wound area was determined at 0, 24, and 96 h using Fiji Software.

### 2.16. MMP Activity

The effect of nanovesicles on MMP activity was determined using the MMP-substrate fluorogenic peptide FS-6 (Mca-Lys-Pro-Leu-Gly-Leu-Dpa-Ala-Arg-NH2) (Sigma-Aldrich, St. Louis, MO, USA) according to a method previously described [33]. Briefly, A549 cells were seeded and incubated with nanovesicles as stated in Section 2.9. After 24 h incubation, 72 μL of cell supernatant was mixed with 8 μL of FS-6 (50 μM) in a 96-well black plate. Fluorescence was followed every 30 min for 6 h at λex = 320 nm and λem = 405 nm in a Cytation™ 5. MMP activity was expressed as the relative fluorescence units (RFU) emitted per hour.

The gelatinolytic activity of MMP-2 and MMP-9 was analyzed by gelatin zymography [34]. Briefly, A549 cells were seeded and incubated with CUR nanovesicles and free CUR as stated in Section 2.9. After 24 h incubation, cell supernatants were collected, concentrated three times by Speed Vac System AES 1010 (Savant, GMI, Inc., Ramsey, MN, USA), and stored at −80 °C until use. Gels were prepared with 10% (*w/v*) polyacrylamide containing 0.1% (*m/v*) SDS and 0.2 mg/mL gelatin from porcine skin type A (Sigma-Aldrich, St. Louis, MO, USA). On each electrophoretic lane, the same amount of total protein supernatants (1000 μg each) was electrophoresed. Electrophoresis was performed using running buffer (0.025 M Tris/0.192 glycine pH 8.3 containing 0.1% *w/v* SDS) at a constant voltage of 120 V. The gels were washed for 1 h in washing buffer (50 mM Tris-HCl, pH 7.4; 5 mM CaCl_2_; 1 μM ZnCl_2_ and 2.5% *w/v* Triton-X 100) to remove SDS. Then, gels were incubated for 48 h at 37 °C in incubation buffer (50 mM Tris-HCl, pH 7.4, 5 mM CaCl_2_, 1 μM ZnCl_2_, 1% *w/v* Triton-X100). After that, gels were stained for 48 h with 0.5% (*w/v*) Coomassie Brilliant Blue R250 to visualize bands of proteolytic activity. Gels were washed out in 5% acetic acid and 10% methanol. Sites of gelatinolytic activity were detectable as clear bands against a background of uniform staining. The intensity of the gelatinolytic bands was measured with Fiji Software. The bands of pro- and active MMP-2 were detected at 72 kDa and 62 kDa and those of pro and active MMP-9 at 92 kDa and 82 kDa, respectively.

### 2.17. Vascular Damage Assay on the CAM

The chicken CAM was used as a model to determine vascular damage produced by photodynamic therapy. Fertilized, specific pathogen-free White Leghorn eggs were supplied by Instituto Rosenbusch S.A. (Buenos Aires, Argentina) on incubation day 7. Undamaged eggs were incubated at 38.0 ± 0.1 °C in a relative humidity of 65 ± 2% and under automatic rotation in an egg incubator (A.Dami Incubadoras, Buenos Aires, Argentina). Between three and six eggs weighing 45–75 g were used to study each condition.

On days 8–9, the air chamber of the eggs was marked and a perforation of 15 mm was made on the shell, then, the internal white membrane was exposed, moistened with 0.9% (*w/v*) NaCl, and removed. A picture of the CAM (time 0) was obtained with a Leica S8APO stereoscopic magnifying glass and recorded with a Leica DMC2900 camera (Leica Microsystem, Wetzlar, Germany). Then, 30 μL of 2 mg/mL solution of mucins from porcine stomach type II (Sigma-Aldrich, St. Louis, MO, USA) was applied on the CAM, immediately after 300 μL of nATC, nLTC, nAT, or nLT at CUR 40 μM and 1 mg/mL of total lipids or PBS were applied over the mucin layer and irradiated with a fluence of 9 J/cm^2^ or kept in the dark (non-irradiated). Control eggs were incubated with PBS and left in the dark or irradiated with a fluence of 9 J/cm^2^ for an irradiation control.

Pictures were obtained at 2, 4, 6, and 8 h after sample incubation and irradiation. The effects on the membrane blood vessels analyzed were lysis and hemorrhage. Lysis was evidenced by the disappearance of small blood vessels, and hemorrhage as bleeding of the vessels on the CAM. Vascular damage parameter results from the sum of the lysis and hemorrhage scores were observed at different times and corrected by a time factor that considers the vascular damage onset.
Vascular damage=∑Score (lysis+haemorrhage)×Time Factor

Lysis, hemorrhage, and vascular damage onset were scored according to the scale described in Appendix A.

### 2.18. Statistical Analysis

Statistical analyses were performed by one-way analysis of variance followed by Dunnet’s test using Prisma 8.0.1 Software. The *p*-value of < 0.05 was considered statistically significant. * *p* < 0.05; ** *p* < 0.01; *** *p* < 0.001; **** *p* < 0.0001; n.s. represents non-significant (*p* > 0.05). Statistical analyses for vascular damage assays were performed by the Kruskal–Wallis non-parametric test followed by Dunn’s multiple comparisons. Differences were considered statistically significant at a *p*-value of < 0.05.

## 3. Results

### 3.1. Spectral Properties and ROS Generation of nATC

The spectral and photochemical properties of CUR are strongly influenced by its keto–enol tautomerism and the formation of intra and intermolecular hydrogen bonds, which depend on the chemical nature of its environment (polarity and protic character) [35]. It was first found that CUR on nATC (size 180 ± 40 nm, ζ potential −24 mV, 150 μg CUR/15 mg lipids/mL, Appendix A) in culture media exhibited maximal fluorescence intensity; in comparison, nLTC (size 226 ± 62 nm, ζ potential −4 mV, 155 μg CUR/15 mg lipids/mL, Appendix A) and free CUR in DMSO diluted in culture media presented less or absent fluorescence (as expected in a protic environment) [14,36]. The absorbance and emission maximums of CUR nanovesicles were shifted to the blue, indicating a more non-polar and less protic environment for CUR, which is optimal to generate ^1^O_2_ [14] (Figure 1a,b). Upon irradiation at CUR absorption maxima λ of ~420 nm with 9 J/cm^2^, the absorbance and fluorescence of CUR formulations significantly decreased as follows: free CUR ~60%, nATC ~50%, and nLTC ~20% (Figure 1), indicating a strong photodegradation of CUR upon irradiation whether formulated in nanoparticles or not, in coincidence with the results reported by Priyadarsini et al. 2009 and Hjorth et al. 1986 [35,37]. CUR is known to produce ROS to a much greater degree than other ortho-methoxyphenols [38]. In culture media, the irradiation of nATC and nLTC, but not of free CUR, induced extracellular ROS, indicating that ROS was more efficiently generated by CUR nanovesicles than by free CUR (Figure 2a). CUR is known to require an aprotic environment to photosensitize ^1^O_2_ formation, while O_2_− generation is reported to take place both in protic and aprotic environments leading to the formation of H_2_O_2_ [14]. The chemical environment of a carried photosensitizer also comprises that of the nanocarrier itself, apart from a given extracellular or intracellular location. In our case, nATC (known to trap CUR for months, whereas liposomes expel CUR to the aqueous media), provided a non-polar environment for CUR; the production of a combination of ^1^O_2_, O_2_−, and H_2_O_2_ is therefore expected.

Upon internalization by A549 cells and irradiated, CUR nanovesicles also strongly induced intracellular ROS (Figure 2b). The generation of intracellular ROS has been linked to the phosphatidylserine externalization and loss of ΔΨm, as early signs of apoptosis, in CUR-treated normal human and human carcinoma cells [39,40]. Accordingly, CUR nanovesicles, but not free CUR, generated micromolar levels of RNS independently from irradiation (Figure 2c).

### 3.2. nATC Mediated PDT on A549 Cells

The photodynamic activity of nATC on A549 cells was determined by measuring the activity of mitochondrial dehydrogenases by the MTT assay. After 1 h incubation with CUR formulations, the cells were irradiated at CUR absorption maxima λ of ~420 nm. It was found first that at >10 μM CUR and 9–18 J/cm^2^ fluences, the cell viability was drastically decreased (>80% decrease) by all formulations (Figure 3). nATC, however, induced the lowest IC_50_ ~3.6 μM CUR (Table 1), nearly half of that of free CUR. Remarkably, despite CUR nanovesicles, TFD was more lethal for A549 cells than when mediated by free CUR, in non-irradiated conditions, the cytotoxicity followed the opposite trend, being free CUR more cytotoxic than in nanovesicles. Irradiation alone, void nanovesicles nAT and nLT, and DMSO did not affect cell viability (Appendix A). It was observed that the cytotoxic effects of PDT were not exclusively owed to CUR internalization since upon washing before irradiation, the cytotoxicity of nATC decreased (IC_50_ 15.3 μM post washing vs. 3.7 μM without washing); the cytotoxicity of nLTC and of free CUR followed a less pronounced similar trend. This suggests that extracellular, non-internalized CUR nanovesicles exerted considerable phototoxicity (Figure 3, Table 1). Together, the observed cytotoxicity on A549 cells resulted from the superposition of photodynamic (favored for CUR nanovesicles, exerted both intra and extracellularly) and chemical effects (favored for free CUR).

### 3.3. PDT-Induced Cell Damage and Cell Death

The assessment of ΔΨm provides a sensitive and early indication of mitochondrial integrity; changes in ΔΨm may be linked to the first stages of programmed cell death. After internalization by A549 cells and irradiation, all formulations reduced ΔΨm to levels comparable to those induced by valinomycin (a potassium ionophore capable of inducing a fast loss of ΔΨm and apoptosis) used as the control (Figure 4a). In the non-irradiated control and above 5 μM CUR, nATC and free CUR induced also ΔΨm depolarization. It can be hypothesized, therefore, that these formulations exerted a predominantly chemical effect on ΔΨm. The increased ΔΨm induced at low CUR concentrations of non-irradiated nLTC can be interpreted as cell proliferation [41].

The decreased production of ATP is directly related to mitochondrial dysfunction and is indicative of cell death. After internalization by A549 cells, irradiated CUR nanovesicles strongly decreased the ATP levels, whereas free CUR induced a less pronounced decrease. In non-irradiated conditions, no significant changes in ATP levels were registered (Figure 4b).

The dyes Annexin V and PI were employed to determine early A549 cell death mechanisms. Once internalized, no signs of cell death were observed 1 h post-irradiation; after 6 h, 40, and 20 μM CUR nATC induced higher levels of early apoptosis (13%) whereas 40 μM free CUR induced higher levels of secondary necrosis (11%) (Figure 5a). At 24 h post-irradiation, all formulations at 20 μM CUR induced a double label of YO-PRO and PI, indicating the cells underwent a necrotic death (Figure 5b). The cell membrane lysis, determined as a necrosis indicator, showed an increased release of LDH with CUR concentration of all formulations 24 h post-irradiation, in coincidence with the YO-PRO and PI levels (Figure 5c). Differences between formulations were observed only during the first hours when nATC induced early apoptosis. Overall, 24 h after internalization and irradiation, all formulations induced necrosis in A549 cells, accompanied by loss of ΔΨm and ATP consumption and LDH release.

### 3.4. Intracellular Uptake of nATC and Lysosomal Damage

The cell uptake of CUR formulations, and further lysosomal location after 1 h incubation, were quantified by fluorescence microscopy by CUR and LR dye fluorescence, respectively. By this means, the lysosomal integrity was also assessed since the LR dye accumulates in healthy lysosomes only. Surprisingly, the amount of intracellular CUR delivered by nATC and free CUR were comparable, and two-fold higher than that delivered by nLTC (Figure 6a,b). Thus, at the moment of irradiation, the cells treated with nATC and free CUR contained a higher amount of intracellular CUR; this may explain the early cell death effects after 1 and 6 h. CUR delivered by nATC moderately colocalized with lysosomes (Rr 0.3) and was homogeneously distributed in the cell cytoplasm (Figure 6a). The fluorescence of LR was 30% higher than in the control cells, indicating an increased lysosomal mass, that may result from the extensive internalization of nATC or the induction of autophagy [42]. CUR delivered by nLTC instead, weakly colocalized with lysosomes, while free CUR markedly accumulated in the cell nuclei. 6 h after irradiation, all formulations induced comparable lysosomal damage (Figure 6c,d). Again, the main differences between formulations were observed during the first hours after internalization; after 24 h, all effects were equalized. The damages induced by photosensitizers on lysosomes are reported to be more lethal than those induced in mitochondria [43,44]. It can be speculated that the early colocalization of nATC in lysosomes induced lysosomal stress and lysosomal membrane permeabilization, cathepsins translocation to the cell cytoplasm and lysosomal-dependent cell death (LDCD)/autophagic cell death [45]. Such events may explain the early apoptosis caused by nATC. This would mean that the structural nature of the nanocarrier, in our case nanoarchaeosomes made of lipids from H tebenquichense, would be more efficient than liposomes to perform intracellular delivery of hydrophobic molecules such as CUR to lysosomes.

### 3.5. nATC Mediated PDT on TAM Models

THP-1 macrophages were cultivated in a conditioned medium from A549 cells previously incubated with nATC and nLTC at 5–40 μM CUR and irradiated. The conditioned media was found to decrease the expression of the CD204 cell membrane receptor on THP-1 macrophages (Figure 7a). Remarkably, 5 μM CUR nATC induced the highest release of IL-6 and was the only one that increased the release of IL-8 (Figure 7b).

### 3.6. Effect of nATC-Mediated PDT on Cell Migration and Anti-MMP Activity

The MMP are molecules needed for tissue invasion as well as a marker of metastasis. The extracellular release of MMP by A549 cells was determined by enzymatic kinetics employing the FS-6 peptide as a fluorogenic substrate. nATC only decreased the MMP release, independently from irradiation, achieving a complete inhibition of MMP at 40 μM CUR. Surprisingly, free CUR stimulated the release of MMP, after cell irradiation, whereas nLTC had no inhibitory activity (Figure 8a). In addition, the gelatin zymography showed that nATC and free CUR, but not nLTC, reduced the degradative activity of pro-MMP-9, activated MMP-9, and activated MMP-2 (Figure 8b). Again, the reduced activity of MMP was independent of irradiation. Void nanovesicles had no inhibitory effect on MMP. The ability to reduce cell migration was determined, by assessing the healing of a wound performed on a 100% confluent A549 cells monolayer. One day after being treated with 5 μM CUR formulations, the wound area remained constant, but four days later, only cells treated with nATC or CUR and irradiated, inhibited cell migration, and impaired the wound healing (Figure 9).

### 3.7. Vascular Damage Induced by PDT

The oxygen reactive species generated by PDT induce irreversible damage in endothelial cells and vascular cell membranes; such damage may collapse the tumoral microvasculature, leading to hemorrhages and tumor destruction by lack of oxygen and nutrients [46]. In this model, the CAM was covered with a mucin layer ~60 μm thick, to mimic the aerial duct’s surface. Poured on the surface, CUR nanovesicles caused significant vascular damage 8 h post-irradiation that was maximal for nLTC. Quantified as a global score, the damage comprised the sum of the lysis and hemorrhage scores observed at different times and corrected by a time factor that considers the vascular damage onset (Figure 10a). Overall, irradiated nATC and nLTC induced shrinkage of the smaller vessels during the first 2 h of incubation, which continued until the complete destruction of the vessels’ net. In the darkness instead, nATC caused no significant vessels harm, the same as void nanovesicles (Figure 10b). The irradiated and non-irradiated control eggs incubated with PBS did not produce any visible change in the vasculature, the same as the Muc control.

## 4. Discussion

CUR is a poorly hydrosoluble and poorly bioavailable drug, and its particulate formulations are mostly designed to increase its oral bioavailability, or, less frequently, to act as a controlled release depot after parenteral administration [6,47]. The design of targeted formulations of CUR aiming to modify its pharmacodynamics has been much less frequently addressed. Targeted formulations often reduce the structural simplicity of nanomedicines, and simplicity is not only a plus but an essential requirement of any scalable nanoparticulate formulation [48]. In this work, we explored the performance of nATC, a structurally stable and naturally targeted nanovesicular formulation of CUR, as a photodynamic agent on A549 cells. Different from liposomal formulations, nATC retains CUR upon dilution and displays a relatively simple targeted structure, a fact that may simplify its scaling up and characterization.

Our results must be analyzed in terms of feasible in vivo future uses, this is the nebulization suitability of each formulation, and the ability to retain CUR over time. Since free CUR cannot be nebulized, and nLTC releases CUR crystals during storage, the present comparison between nATC and nLTC and free CUR was performed only to assess the role of the archaeolipid matrix and to compare the effects of nanovesicles vs. those of free CUR after cell uptake. In such context, it was found that the archaeolipid matrix of nATC did not exert any additional cyto or phototoxicity compared to the liposomal matrix; after 24 h of irradiated, free CUR, nLTC, and nATC caused the same lysosomal damage, and a decrease in ΔΨm, induced the same extent of ATP consumption, whereas the extra and intracellular generation of ROS induced by 40 or 20 μM of CUR nanovesicles, was practically the same. The cytotoxicity of non-irradiated formulations followed the trend: CUR> nATC> nLTC; upon irradiation at 9 J/cm^2^, the order of damage was modified and the IC_50_ values were substantially reduced by 30-fold for nATC and nLTC, resulting in the nanovesicles being more cytotoxic than free CUR. This signified that nanovesicles, independently of their chemical nature, would favor the exertion of intracellular phototoxicity. This may be related to the higher ability of CUR nanovesicles to produce ROS after irradiation and RNS in the dark. The only effect that could be ascribed solely to the archaeolipids matrix was the fact that non-irradiated, 40 μM CUR nATC impaired the production of MMP-2 and MMP-9 by A549 cells. This chemical effect is in line with the previously reported ability of nATC in repairing epithelia [49].

Apart from A549 phototoxicity, two remarkable effects of nATC were caused. One was that upon internalization of 5 uM nATC and irradiation, their migration was impaired. The other was that conditioned media from A549 cells treated with 5 μM nATC and irradiated reduced the expression of the CD204 marker on macrophages derived from THP-1 monocytes. Used as models for TAM, the reduced expression of CD204 suggests a reduction of their pro-tumoral M2 phenotype. This is important since the tumoral microenvironment of epithelial tumors such as NSCLC predominantly contains M2 phenotypic TAM [50], which is known to express high SRA1 levels, a receptor responsible for the extensive internalization of nATC [51]. Hence, besides direct antitumoral activity, the irradiation of these tumors may reprogram their TAM to an antitumor phenotype, as judged by the increased production of pro-inflammatory cytokines that followed the treatment with nATC.

CAM has been previously used to model vessels submitted to CUR-PDT. The CAM surface received a high CUR concentration (100 μM CUR in saline solution 0.1% ethanol) and was irradiated at high fluence (30 J/cm^2^), which first induced shrinkage and ended in a complete collapse of small vessels after 30 min [52].

In our approach, the CAM was used to model vessels under a mucin layer covering the CAM surface, which was additionally covered with a mucin layer ~0.06 mm thick (comparable thickness to the 5–50 μm thick mucus layer covering the epithelia of lung bronchia) [53]. After 8 h and at 40 μM, irradiated CUR nanovesicles induced the shrinking and further collapse of the vessels. These results suggest therefore that in vivo, the mucin layer on the lung airways would not impair the access of nebulized vesicles to the subjacent epithelia. This also showed that upon topical administration of CUR formulations and blue light irradiation, the generation of phototoxicity at low depth (capillary plexus is located immediately below the chorionic epithelium, the outer cell layer of the CAM [54]) from a surface was possible. Remarkably, different from nLTC, non-irradiated nATC was harmless, even after 8 h. Therefore, while the vascular damage caused by nATC was limited to irradiated zones, that of nLTC would escape from the light control.

In addition, recently, the group of Bakowski explored the performance of nebulized CUR nanovesicles to perform antitumoral PDT, consisting of bilayers lacking Tween 80 and containing 90% weight DPPC (synthetic hydrogenated ordinary phospholipid) and 10% TEL (tetraether from the archaea *Sulfolobus acidocaldarius*) [23]. *S. acidocaldarius* is a thermoacidophile microorganism thriving in extreme conditions of temperature (80 °C) and acidity (pH < 2); its lipids are highly thermostable caldarchaeols containing pentacycles, that form monolayers [55]. In there, the use of CAM was performed classically, without mucins and determining the damage induced 5 min after nebulization.

Because of their simpler growth conditions and lower costs, halophilic archaea are of the outmost biotechnological potential as sources of polar and non-polar archaeolipids [56]. Importantly, the biotechnological use of archaea products is in its beginning stage. The polar archaeolipids possess a biotechnological readiness level 3 [57]. This signifies that archaeolipids are products whose demand is satisfied by tiny amounts commercialized at high prices. In such context, archaeosomes from halophilic archaea constitute novel structural platforms for drug delivery, alternative to liposomes. Besides strongly trapping poorly hydrosoluble drugs, these archaeosomes are less refractory to oxidation and hydrolytic attacks than those made of tetraethers, but strong enough to maintain their colloidal structure against realistic aggressive environments. As proof of the versatility of such structural platform, recently void archaeosomes made of halophilic dieters were revealed to be bioactive, displaying an intense anti-inflammatory activity on vascular endothelial cells [58]. Remarkably, the reduced viability and migration of A549 cells, the vessel lysis, the reduced expression of CD204 by THP1-derived macrophages (as TAM models), or the chemical reduction of MMP release and pro-inflammatory cytokines previously reported [22] could be exerted by nATC between a short range of concentrations of 5–40 μM CUR. Our results suggest therefore that the activity of nATC occurred at multiple levels, that can be triggered and controlled by a blue light of low fluency according to the specific need.

## Figures and Tables

**Figure 1 pharmaceutics-14-02637-f001:**
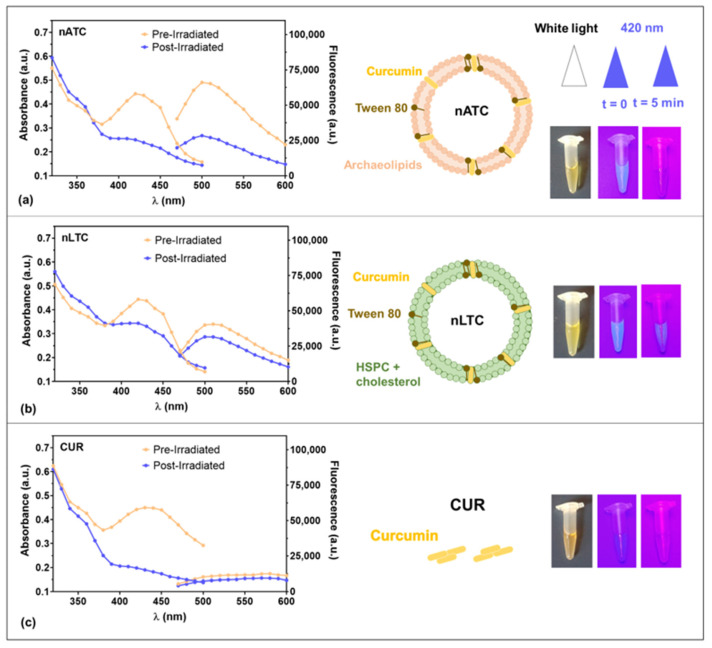
Absorbance and fluorescence spectra of 40 μM (**a**) nATC, (**b**) nLTC, or (**c**) free CUR, before and after irradiation, with a fluence of 9 J/cm^2^.

**Figure 2 pharmaceutics-14-02637-f002:**
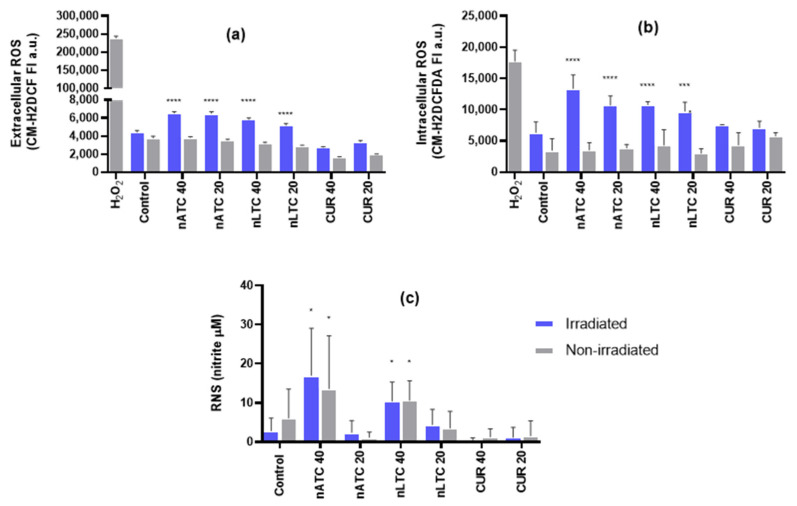
ROS and RNS production by PDT. (**a**) ROS production by irradiated (fluence of 9 J/cm^2^) or non-irradiated nATC, nLTC and free CUR, suspended in culture medium (RPMI 5% of FBS). (**b**) Intracellular ROS and (**c**) RNS production by A549 cell line after 24 h incubation with irradiated (fluence of 9 J/cm^2^) or non-irradiated nATC, nLTC or free CUR. Statistical significance compared to non-irradiated control was determined using a one-way ANOVA followed by Dunnet’s test, * *p* < 0.05; *** *p* < 0.001, **** *p* < 0.0001.

**Figure 3 pharmaceutics-14-02637-f003:**
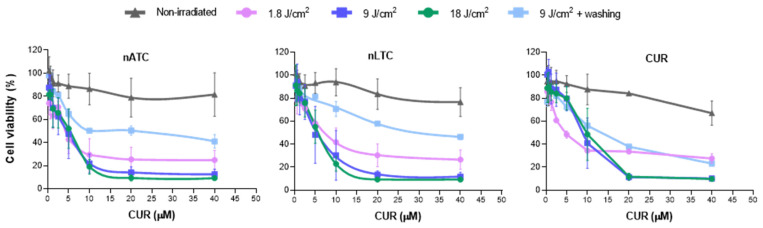
PDT on A549 cell line. Cells were incubated with nATC, nLTC, or free CUR and irradiated with a fluence of 1.8, 9, and 18 J/cm^2^ or kept in the dark (non-irradiated). In addition, cells were incubated with formulations, washed with PBS, and irradiated with a fluence of 9 J/cm^2^. Cell viability was determined after 24 h incubation by MTT.

**Figure 4 pharmaceutics-14-02637-f004:**
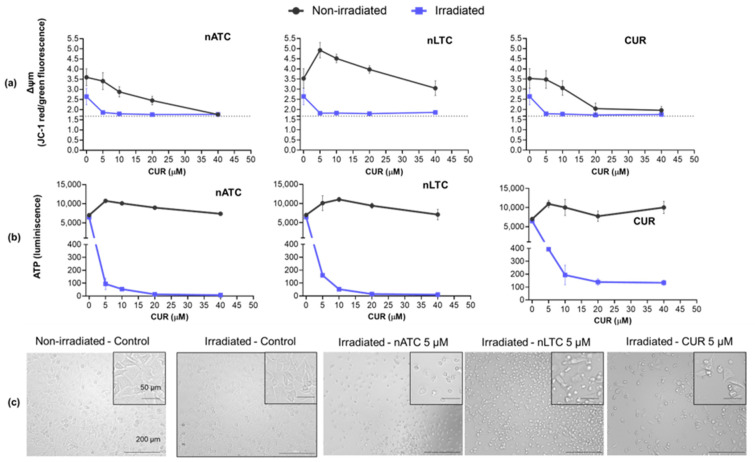
PDT cell damages on A549 cell line. Cells were incubated with nATC, nLTC, or free CUR at 5, 10, 20, and 40 μM, irradiated with a fluence of 9 J/cm^2^ or kept in the dark (non-irradiated). After 24 h, changes in (**a**) mitochondrial membrane potential (ΔΨm) and (**b**) ATP content were measured. (**c**) Bright-field microscopy images of A549 cell line after PDT, with 5 μM CUR and a fluence of 9 J/cm^2^.

**Figure 5 pharmaceutics-14-02637-f005:**
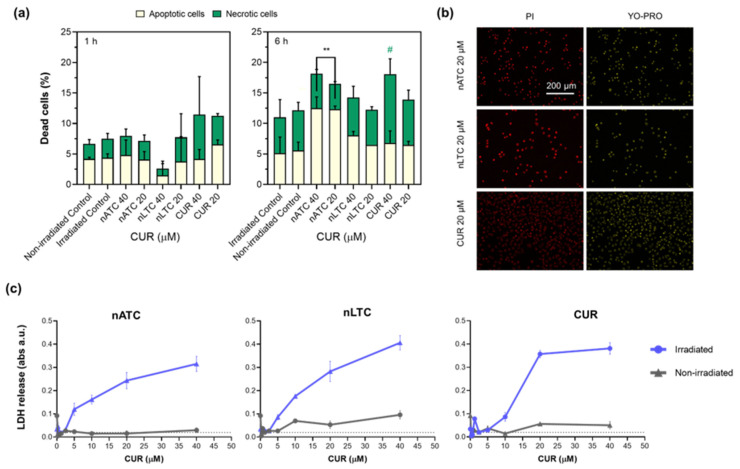
Death mechanisms induced by PDT. A549 cell line was incubated with nATC, nLTC, or free CUR and irradiated (fluence of 9 J/cm^2^) or kept in the dark (non-irradiated). (**a**) 1 and 6 h after irradiation cells were incubated with Annexin V and PI. Significant differences compared with apoptotic control cells were determined using a one-way ANOVA followed by Dunnet’s test, ** *p* < 0.01. Significant differences compared with necrotic non-irradiated control cells were determined using a one-way ANOVA followed by Dunnet’s test, # *p* < 0.05. After 24 h incubation (**b**), cells were stained with YO-PRO and IP and (**c**) LDH release in cells supernatants was measured.

**Figure 6 pharmaceutics-14-02637-f006:**
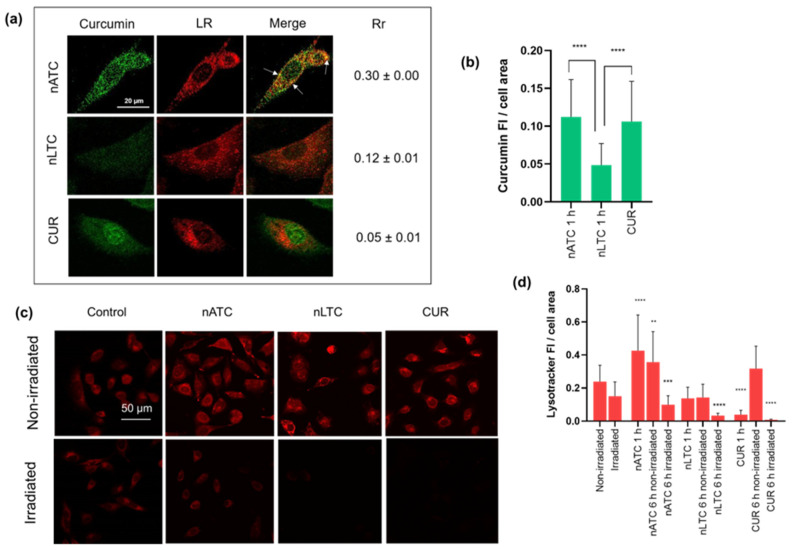
A549 cellular uptake of CUR and lysosomal damage. (**a**) Confocal microscopy images of non-irradiated nATC, nLTC, and CUR after 1 incubation. Green dots represent CUR, red dots represent Lysotracker Red (LR). Pearson’s coefficient (Rr) at 1 h incubation. White arrows indicate yellow dots of colocalization. (**b**) CUR fluorescence intensity inside the cells after 1 h incubation. Significant differences between formulations were determined using a one-way ANOVA followed by Dunnet’s test, **** *p* < 0.0001. (**c**) Fluorescence images of LR after 6 h incubation with irradiated or non-irradiated nATC, nLTC, and CUR. (**d**) Lysotracker Red (LR) fluorescence intensity inside the cells after 6 h incubation. Significant differences between formulations were determined using a one-way ANOVA followed by Dunnet’s test, ** *p* < 0.01; *** *p* < 0.001; **** *p* < 0.0001.

**Figure 7 pharmaceutics-14-02637-f007:**
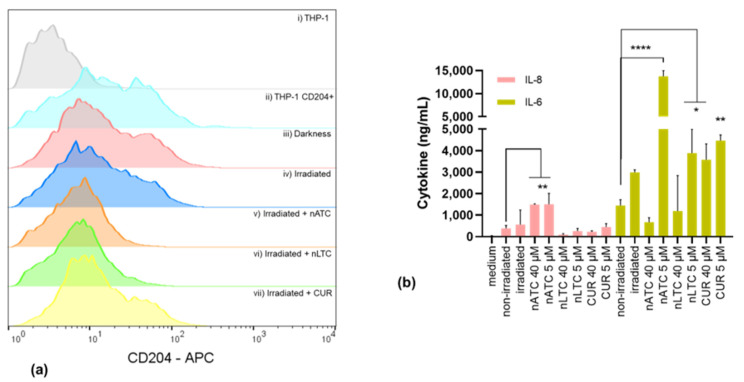
Tumor-associated macrophages (TAM) modulation. (**a**) Expression of CD204 in THP-1 cells line transformed to macrophages, (i) THP-1 control, (ii) THP-1 with CD204 labeling, (iii) THP-1 after 24 h incubation with supernatants obtained from non-irradiated A549 cells, (iv) THP-1 after 24 h incubation with supernatants obtained from A549 cells irradiated with a fluence of 9 J/cm^2^, (v) THP-1 after 24 h incubation with supernatants of A549 previously incubated with 5 μM nATC, (vi) 5 μM nLTC and (vii) 5 μM CUR and irradiated. (**b**) IL-8 and IL-6 quantification in supernatants of THP-1 cells, cells without treatment (Medium), after 24 h incubation with supernatants of irradiated A549, or non-irradiated A549, or A549 incubated with 5 μM or 40 μM nATC, nLTC, or free CUR and irradiated. Statistical significance compared with non-irradiated control was determined using a one-way ANOVA followed by Dunnet’s test, * *p* < 0.05; ** *p* < 0.01, **** *p* < 0.0001.

**Figure 8 pharmaceutics-14-02637-f008:**
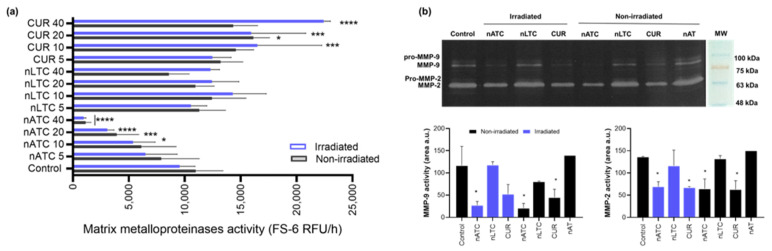
Effect of PDT on matrix metalloproteinases (MMP). MMP activity was measured on A549 cell supernatants 24 h after incubation with nATC, nLTC, and free CUR irradiated (fluence of 9 J/cm^2^) or kept in the dark (non-irradiated). MMP activity was determined using (**a**) the fluorogenic peptide FS-6 for general MMP activity. Statistical significance compared to non-irradiated control was determined using a one-way ANOVA followed by Dunnet’s test, * *p* < 0.05; *** *p* < 0.001, **** *p* < 0.0001. (**b**) Gelatin zymography for specific MMP-2 and MMP-9 collagenase activity. Significant differences in gel degradation areas compared to non-irradiated control cells were determined using a one-way ANOVA followed by Dunnet’s test, * *p* < 0.05.

**Figure 9 pharmaceutics-14-02637-f009:**
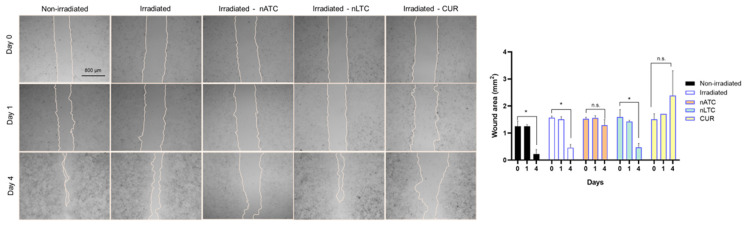
Effect of PDT on cell migration. A549 cells were grown to total confluence and a wound was produced with a tip (day 0). Then cells were treated with nATC, nLTC, or CUR at 5 μM and irradiated (fluence of 9 J/cm^2^). After 24 h, cells were washed with PBS (day 1) and fresh medium was added, then cells were incubated for another 72 h (day 4). Images were obtained with bright filed microscopy at Days 0, 1, and 4. Significant differences in wound area compared with non-irradiated control were determined using a one-way ANOVA followed by Dunnet’s test, * *p* < 0.05; n.s. represents non-significant (*p* > 0.05).

**Figure 10 pharmaceutics-14-02637-f010:**
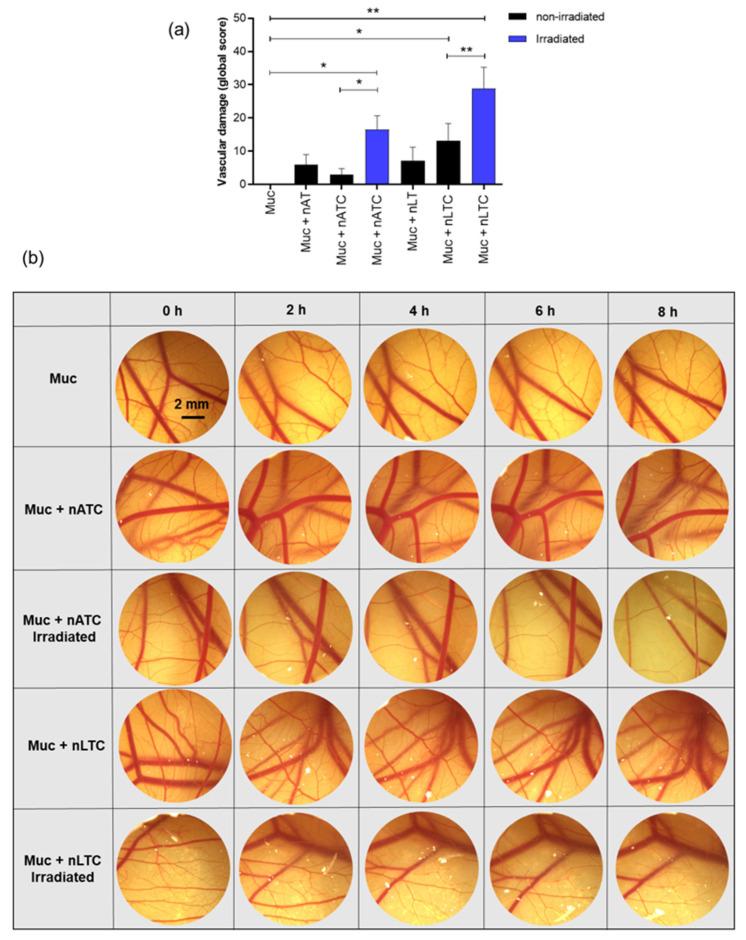
Vascular damage induced by nATC, nLTC, and free CUR on the CAM covered with mucins (Muc). (**a**) The global score of accumulated vascular damage after 8 h post-incubation with nATC, nLTC, or free CUR and irradiation with a fluence of 9 J/cm^2^ or kept in the dark (non-irradiated). Statistical analyses were performed by Kruskal–Wallis non-parametric test followed by Dunn’s multiple comparisons. Differences were considered statistically significant at *p* < 0.05, * *p* = 0.0174, ** *p* = 0.0078. (**b**) Images of CAM vasculature at different time points after sample incubation and irradiation with a fluence of 9 J/cm^2^ or kept in the dark (non-irradiated).

**Table 1 pharmaceutics-14-02637-t001:** IC_50_ values (μM) on A549 cell line with 1.8, 9 and 18 J/cm^2^ fluences ^1^.

Formulation	Non-Irradiated	1.8 J/cm^2^	9 J/cm^2^	18 J/cm^2^	Washing + 9 J/cm^2^
nATC	121 ± 39	3.6 ± 1.2	3.7 ± 0.5	3.6 ± 0.6	15.3 ± 4.5
nLTC	163 ± 43	7.1 ± 1.3	4.9 ± 0.8	4.9 ± 0.7	26.2 ± 6.3
CUR	83 ± 22	4.07 ± 0.8	8.5 ± 1.6	8.9 ± 1.7	12.2 ± 3.1

^1^ IC_50_ values were obtained after 24 h incubation post-irradiation by MTT assay.

## Data Availability

The data presented in this study are available on request from the corresponding author.

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
