# Peer review of "Photodynamic Therapy with Nebulized Nanocurcumin on A549 Cells, Model Vessels, Macrophages and Beyond"

_pharmaceutics, 2022, doi:10.3390/pharmaceutics14122637_

Round 1

Reviewer 1 Report

The article demonstrates the advantages of nebulized nanocurcumin (nATC) over nLTC in the aspect of potency and toxicity. The experiments were well designs. Some minor points should be considered. 

1. The relationship between properties and antitumor activity outcome. For example, nATC is located in higher concentration in the lysosome, is this property related to the potency of nATC (such as cell damage and cell death)?  

2. Please discuss the limitation of this study and the application in vivo. The study demonstrated that ROS and RNS generated in both extra- and intra-cellular. Therefore, the toxicity of cur formulation nATC may not specific at the tumor site if the rate of cellular uptake is limited.

3. Please consider modifying the Figure axis. For example, in figure 1-7 the order of the studied group is nATC-nLTC-Cur, but in figure 8b Cur-nLTC-nATC.  As well as figure 9, the image and bar graph is not consistent.    

4. Please check the label of Figure 5 with the description in the main text and the symbol of figure 5c.    

5. Please check significant of muc+nLTC dark and irradiated in Figure 10a.

Author Response

The article demonstrates the advantages of nebulized nanocurcumin (nATC) over nLTC in the aspect of potency and toxicity. The experiments were well designs. Some minor points should be considered.

Point 1. The relationship between properties and antitumor activity outcome. For example, nATC is located in higher concentration in the lysosome, is this property related to the potency of nATC (such as cell damage and cell death)?

Answer: Thank you for pointing out these subjects. We have included part of the following paragraph in the Discussion:

The damages induced by photosensitizers on lysosomes are reported to be more lethal than those induced in mitochondria  [Enhanced efficiency of cell death by lysosome-specifc photodamage Tayana Mazin Tsubone, Waleska Kerllen Martins, Christiane Pavani, Helena Couto Junqueira, Rosangela Itri & Maurício S. Baptista. SCIeNtIFIC Reports | 7: 6734 ] [Kessel, D. Autophagic death probed by photodynamic therapy. Autophagy 11, 1941–1943 (2015)]. It can be speculated that the early co-localization of nATC in lysosomes induced lysosomal stress  and lysosomal membrane permeabilization, cathepsins translocation to the cell cytoplasm and lysosomal-dependent cell death (LDCD)/autophagic cell death [Lysosomal membrane permeabilization and cell deathFengjuan Wang, Raquel Gómez-Sintes,  Patricia Boya. Traffic. 2018;19:918–931]. Such events may explain the early apoptosis caused by nATC  (see 3.3 PDT-induced cell damage and cell death). This would mean that the structural nature of the nanocarrier-in our case nanoarchaeosomes made of lipids from  H tebenquichense-,  would be more efficient that liposomes to perform intracellular delivery of hydrophobic molecules such as CUR , to lysosomes.  

Point 2. Please discuss the limitation of this study and the application in vivo. The study demonstrated that ROS and RNS generated in both extra- and intra-cellular. Therefore, the toxicity of cur formulation nATC may not specific at the tumor site if the rate of cellular uptake is limited.

Answer: Thank you for your comment. It is true that the generation of ROS and RNS occurred both intra and extracellularly. However, both were only produced upon irradiation, meaning that ROS and RNS would be limited to the places under illumination. In addition, the lifetime of the highly reactive 1O2 (one of the ROS specials that may be produced by nATC-mediated PDT), oscillates between 10-100 us (2 us in aqueous media); its activity, therefore, would be limited to a sphere of 10 nm diameter, with low possibilities of diffusing [Basic principles of photodynamic therapy. IAN J. MACDONALD* and THOMAS J. DOUGHERTY J. Porphyrins Phthalocyanines 2001; 5: 105–129], [Basic principles of photodynamic therapy. IAN J. MACDONALD* and THOMAS J. DOUGHERTY J. Porphyrins Phthalocyanines 2001; 5: 105–129]. nATC on the other hand, is a vesicular formulation that strongly traps  CUR for months. It could be speculated therefore, that unless nATC is internalized, at least part of the generated ROS would remain mostly on the vicinities of the nATC bilayer.

Point 3. Please consider modifying the Figure axis. For example, in figure 1-7 the order of the studied group is nATC-nLTC-Cur, but in figure 8b Cur-nLTC-nATC. As well as figure 9, the image and bar graph is not consistent.

Answer: Thank you for the comment, we unified the order of the samples in all figures.

Point 4. Please check the label of Figure 5 with the description in the main text and the symbol of figure 5c.

Answer: Thank you for the comment, we corrected the label of Figure 5 and the description in the main text.  

Point 5. Please check significant of muc+nLTC dark and irradiated in Figure 10a.

Answer: We analyzed the results of Figure 10.a with Kruskal-Wallis non-parametric test followed by Dunn’s multiple comparisons to determine differences between muc+nLTC dark (non-irradiated) and irradiated. We repeated statistical analysis and confirmed that there is no significant difference between treatments.

Reviewer 2 Report

In this work, Maria developed curcumin nanoparticles by encapsulating curcumin in nanovesicles for photodynamic therapy. The nanocurcumin were characterized with capacity to cause damage on A549 cells, induce polarization of macrophages, and disrupt superficial vasculatures, showing potential for clinical translations. However, it’s not recommended for publications in Pharmaceutics for some considerations.

1.     There is no characterization on the size or morphology of nanocurcumin through the manuscript. Transmission electron microscopy, scan electron microscopy, or dynamic light scattering characterization of nanocurcumin must be provided.

2.     The novelty of this work is diminished since curcumin nanoparticles have been widely reported for photodynamic therapy.

3.     Blue light irradiation is with short wavelength. Whether it would cause phototoxicity to nor tissues when combines administration with nanocurcumin?

4.     Though curcumin is approved by FDA, what’s the advantage of curcumin over some photosensitizers longer-wavelength absorption, such as derivatives of porphyrin, BODIPYs?

5.     Scale bars should be added to Figure 5b, Figure 6c, and Figure 10b.

6.     On day 0 of Figure 9, the scratch width of Irradiated-CUR seemed lager than control group.

7.     Cellular reactive oxygen species (ROS) content induced by nanocurcumin should be evaluated by using DCFH-DA.

8.     What’s the singlet oxygen quantum yield of curcumin?

9.     What kind of ROS does nanocurcumin generate?

10.  The photophysical mechanisms of PDT should be briefly introduced in introduction section.

Author Response

In this work, Maria developed curcumin nanoparticles by encapsulating curcumin in nanovesicles for photodynamic therapy. The nanocurcumin were characterized with capacity to cause damage on A549 cells, induce polarization of macrophages, and disrupt superficial vasculatures, showing potential for clinical translations. However, it’s not recommended for publications in Pharmaceutics for some considerations.

Point 1. There is no characterization on the size or morphology of nanocurcumin through the manuscript. Transmission electron microscopy, scan electron microscopy, or dynamic light scattering characterization of nanocurcumin must be provided.

Answer: As we report in the Introduction (line 81) the design and physicochemical characterization of nATC and nLTC was presented in a previous research paper of our group (Altube, M.J.; Caimi, L.I.; Huck-Iriart, C.; Morilla, M.J.; Romero, E.L. Reparation of an Inflamed Air-Liquid Interface Cultured A549 Cells with Nebulized Nanocurcumin. Pharmaceutics 2021, 13, doi:10.3390/pharmaceutics13091331.) We determined nanovesicles size, Z potential and morphology by DLS and Cryo-electron microscopy. We also studied de interaction of curcumin molecules with de lipid bilayer of nATC and nLTC by Laurdan generalized polarization and fluorescence anisotropy, SAXS, RAMAN, absorbance and fluorescence spectroscopy. In addition, we determined the stability of the formulations upon nebulization and after six months storage.

For clarity, we included in the main text the results of Z average, Z potential and CUR-loading capacity in nanovesicles. These results are an average of every determination that we rutinary evaluated in every newly batch of nATC and nLTC that we prepared. In addition, we included a Table with these results in Supplementary information (Table S2.)

Point 2. The novelty of this work is diminished since curcumin nanoparticles have been widely reported for photodynamic therapy.

Answer: We appreciate the reviewer´s comment. However, most uses of CUR-based PDT are focused on antimicrobial and much less on its antitumoral activity [Curcumin as a photosensitizer: From molecular structure to recent advances in antimicrobial photodynamic therapy Lucas D. Diasa,∗, Kate C. Blancoa, Ivan S. Mfouo-Tyngaa, Natalia M. Inadaa, Vanderlei S. Bagnatoa Journal of Photochemistry and Photobiology C: Photochemistry Reviews 45 (2020) 100384].

Point 3. Blue light irradiation is with short wavelength. Whether it would cause phototoxicity to nor tissues when combines administration with nanocurcumin?

Answer: Our controls showed that the irradiation with blue LEDs at 9 J/cm2 did not cause any harm to A549 or THP-1 derived macrophage cell cultures. Besides, the energy densities of blue lasers employed in dermatology are 5-50 times higher  [Cios, A.; Ciepielak, M.; Szyma ´nski, Ł.; Lewicka, A.; Cierniak, S.; Stankiewicz, W.; Mendrycka, M.; Lewicki, S. Effect of Different Wavelengths of Laser Irradiation on the Skin Cells. Int. J. Mol. Sci. 2021, 22, 2437. https://doi.org/10.3390/ ijms22052437]

Point 4. Though curcumin is approved by FDA, what’s the advantage of curcumin over some photosensitizers longer-wavelength absorption, such as derivatives of porphyrin, BODIPYs?

Answer: It is true that a disadvantage of CUR is its low wavelength absorption maximum.  However, the exploration of its use as photosensitizer is related with its low cytotoxicity for healthy cells, and its ability to provide anti-inflammatory, antioxidant, antimicrobial and antitumoral activities in the absence of irradiation (Curcumin encapsulation in nanostructures for cancer therapy: A 10-year overview Nat´alia A. D’Angelo a, Mariana A. Noronha a, Isabelle S. Kurnik b, Mayra C.C. Cˆamara a, Jorge M. Vieira c, Luís Abrunhosa c, Joana T. Martins c, Thais F.R. Alves d,e,f, Louise L. Tundisi a, Janaína A. Ataide a, Juliana S.R. Costa a, Angela F. Jozala g, Laura O. Nascimento a, Priscila G. Mazzola a, Marco V. Chaud d,e,f, Ant´onio A. Vicente c, Andr´e M. LopesInternational Journal of Pharmaceutics 604 (2021) 120534). Moreover, it has been hypothesized that CUR displays a dual activity, as free radical scavenger or producer, acting as  antioxidant in normal cells and prooxidant in tumor cells. [B.B. Aggarwal , B. Sung , Pharmacological basis for the role of curcumin in chronic diseases: an age-old spice with modern targets, Trends Pharmacol. Sci. 30 (2009) 85–94].

As indicated in Introduction, the FDA has approved the use of curcumin as food colour additive   [https://www.accessdata.fda.gov/scripts/cdrh/cfdocs/cfcfr/CFRSearch.cfm?fr=73.600], but not as therapeutic agent.

Point 5.     Scale bars should be added to Figure 5b, Figure 6c, and Figure 10b.

Answer: Thank you for the comment, we included scale bars in Figure 5b, Figure 6c and Figure 10b.

Point 6. On day 0 of Figure 9, the scratch width of Irradiated-CUR seemed lager than control group.

Answer: The mean area on Day 0 of Control (non-irradiated) was 1.25 ± 0.14 mm2 and for CUR 1.5 ± 0.15 mm2, we did not determine significant differences in the initial area between treatments. In addition, the comparisons of wound healing were made between Days of the same treatment.

Point 7. Cellular reactive oxygen species (ROS) content induced by nano curcumin should be evaluated by using DCFH-DA.

Answer: In Figure 2.b we presented the results of ROS on A549 cells induced by nATC, nLTC and free CUR, also a positive control of H2O2 is included. The determination was performed with the probe chloromethyl-2',7'-dichlorodihydrofluorescein diacetate (CM-H2DCFDA) from Thermo Fisher Scientific, Waltham, MA, USA. We modified y-axis of Figure 2 and include the name of the probe for clarity

Point 8. What’s the singlet oxygen quantum yield of curcumin?

Answer: The singlet oxygen quantum yield (ɸ1O2) of  CUR is low and depends on the nature of its chemical environment: 0.12 in deuterated benzene [A.A. Gorman, I. Hamblett, V.S. Srinavasan, P.D. Wood, Photochem. Photobiol.59 (1994) 389–398.], 0.11 in toluene and acetonitrile  [C.F. Chignell, P. Bilski, K.J. Reszka, A.N. Motten, R.H. Sik, T.A. Dhal, Photochem. Photobiol. 59 (1994) 295–302.] whereas in ethanol, isopropanol, SDS and TX-100 micelles in, D2O ɸ1O2 <0.005.

Point 9. What kind of ROS does nanocurcumin generate?  

Answer: To answer the reviewer´s query, the following paragraphs have been partly included in the section “3.1 Spectral properties and ROS generation of nATC”: Curcumin is known to require an aprotic environment to photosensitize 1O2 formation, while O2·− generation is reported to take place both in protic and aprotic environment leading to the formation of H2O2 ([C.F. Chignell, P. Bilski, K.J. Reszka, A.N. Motten, R.H. Sik, T.A. Dhal, Photochem. Photobiol. 59 (1994) 295–302.]). The chemical environment of a carried photosensitizer comprises also that provided by the nanocarrier itself, apart from a given extracellular or intracellular location.   In our case, nATC ( known to trap CUR for months -whereas liposomes expel CUR to the aqueous media-), provided a less hydrophilic environment for CUR; the production of a combination of 1O2, O2·−  and H2O2 is therefore expected.  On the other hand, studies on PDT with curcumin in bacteria and tumor cells that studied the generation of different reactive species such as H2O2, 1O2, O2·−, observed that most of the total cellular phototoxicity resulted from curcumin reactive intermediates formed in the extracellular medium rather than from the association of curcumin with cells and the production of ROS. (Bruzell, E. M., Morisbak, E., & Tønnesen, H. H. (2005). Studies on curcumin and curcuminoids. XXIX. Photoinduced cytotoxicity of curcumin in selected aqueous preparations. Photochemical and Photobiological Sciences, 4(7), 523–530; Dahl’, T. A., Bilski~, P., Reszka~, K. J., & Chignell, C. F. (1994). PHOTOCYTOTOXICITY OF CURCUMIN (Vol. 59, Issue 3), Dahl, T. A., Mcgowan, W. M., Shand, M. A., & Srinivasan, V. S. (1989). Photokilling of bacteria by the natural dye curcumin*. In Arch Microbiol (Vol. 151).

Point 10. The photophysical mechanisms of PDT should be briefly introduced in introduction section.

Answer: Thank you for the suggestion. The following paragraph: “The photodynamic mechanism is based on the interaction between the excited photosensitizer and surrounding oxygen molecules, generating reactive oxygen species, particularly singlet oxygen (1O2) [7]” has been included in the Introduction.

Reviewer 3 Report

In the current manuscript, the authors presented the PDT activity of curcumin against cancerous cells. The work is interesting, however, the presentation of results needs to be revised up to the mark. I would recommend a revision before the publication of the manuscript. Some of my specific comments are below:

First I would suggest a little change in the title, Authors wrote the “antitumoral activity” however, in this work no tumor model is characterized thus I would suggest removing this word and using any other alternative one.

Rewrite the abstract to specifically elaborate Problem, its solution, the method utilized, the results obtained, and the conclusion.

The manuscript needs to be carefully edited by a native English speaker. Each section should be carefully revised.

All short names must be abbreviated at their first appearance. See line 16, an abbreviation of nATC? Line 21, ROS, etc.

Line 17: Change the symbol of Zeta potential (ζ)

Figure 1: It is hard to see yellow-colored naming such as curcumin. Increase the font size and make them bold. Increase the caption fonts in graphs a, b, and c, it’s hard to see “pre and post-irradiated” captions even after increasing the page size by 125%.

Figure 2: Increase the quality by increasing the column width a little bit. The column caption is written as “Irradiated and darkness” change them to “Irradiated (420 nm), and non-irradiated (Dark)”, respectively.

Elaborate lines 411-412 “indicating a strong photodegradation of CUR upon irradiation whether formulated in nanoparticles or not, in coincidence with????” is hard to understand.

If the light source is in-house incorporate a picture of LED light with proper setup. If authors used a procured PDT laser incorporates the name of the manufacturer.

Rewrite the sentence from 436-438, “It was found first that at >10 μM CUR and 9–18 J/cm2 fluences, all formulations drastically decreased (>80% decrease) the cells viability (Figure 3).” Formulation drastically decreased meaning? Here the cell viability is supposed to be decreased, right?

In the next sentence, author wrote, “nATC induced the lowest IC50 ~3.6 μM CUR (Table 1), doubling that of free CUR.” What is the meaning of this sentence? Induced the lowest IC50????? Rewrite properly. I would suggest rewriting the whole paragraph properly.

In the figure wherever you have done PDT comparisons do not write “darkness” Instead of that write “non-irradiated”

In figure 3 or other PDT studies the effect of light only also needs to be added. Such as in figure 2, the effect of light without treating with the formulations should be added.

Figure 4, the line captions in the graphical figures a and b are not given. What blue and black lines are represented here?

In each figure authors used ANOVA analysis, however, they have not mentioned any Post-Hoc test. What post-hoc test was applied for comparing the groups?

Author Response

In the current manuscript, the authors presented the PDT activity of curcumin against cancerous cells. The work is interesting, however, the presentation of results needs to be revised up to the mark. I would recommend a revision before the publication of the manuscript. Some of my specific comments are below:

Point 1. First I would suggest a little change in the title, Authors wrote the “antitumoral activity” however, in this work no tumor model is characterized thus I would suggest removing this word and using any other alternative one.

Answer: The title has been modified according to the reviewer´s suggestion.

Point 2. Rewrite the abstract to specifically elaborate Problem, its solution, the method utilized, the results obtained, and the conclusion.

Answer: The abstract has been re-written according to the reviewer´s suggestion

Point 3. The manuscript needs to be carefully edited by a native English speaker. Each section should be carefully revised.

Answer: The manuscript has been re-edited.

Point 4. All short names must be abbreviated at their first appearance. See line 16, an abbreviation of nATC? Line 21, ROS, etc.

Answer: Thank you for the comment, we modified the abbreviations 

Point 5. Line 17: Change the symbol of Zeta potential (ζ)

Answer: Thank you for the comment, we changed the symbol of Zeta potential

Point 6. Figure 1: It is hard to see yellow-colored naming such as curcumin. Increase the font size and make them bold. Increase the caption fonts in graphs a, b, and c, it’s hard to see “pre and post-irradiated” captions even after increasing the page size by 125%.

Answer: Thank you for the comment, we modified Figure 1.

Point 7. Figure 2: Increase the quality by increasing the column width a little bit. The column caption is written as “Irradiated and darkness” change them to “Irradiated (420 nm), and non-irradiated (Dark)”, respectively.

Answer: Thank you for the comment, we modified Figure 2.

Point 8. Elaborate lines 411-412 “indicating a strong photodegradation of CUR upon irradiation whether formulated in nanoparticles or not, in coincidence with????” is hard to understand.

Answer: The sentence has been re-written

Point 9. If the light source is in-house incorporate a picture of LED light with proper setup. If authors used a procured PDT laser incorporates the name of the manufacturer.

Answer: We included an image of the home-made device in Supplementary information Figure S1

Point 10. Rewrite the sentence from 436-438, “It was found first that at >10 μM CUR and 9–18 J/cm2 fluences, all formulations drastically decreased (>80% decrease) the cells viability (Figure 3).” Formulation drastically decreased meaning? Here the cell viability is supposed to be decreased, right?

Answer: The sentence has been re-written

Point 11. In the next sentence, author wrote, “nATC induced the lowest IC50 ~3.6 μM CUR (Table 1), doubling that of free CUR.” What is the meaning of this sentence? Induced the lowest IC50????? Rewrite properly. I would suggest rewriting the whole paragraph properly.

Answer: The sentence has been re-written

Point 12. In the figure wherever you have done PDT comparisons write “darkness” Instead of that write “non-irradiated”

Answer: Thank you for the comment, we modified “darkness” for “non-irradiated” in all the figures and in the main text.

Point 13. In figure 3 or other PDT studies the effect of light only also needs to be added. Such as in figure 2, the effect of light without treating with the formulations should be added.

Answer: Thank you for the comment, we made the following modifications:

Figure 3: Irradiated Control at different fluences was included in Supplementary Information Figure S1 and it was mentioned in line 461 of the main text.

Figure 4c: we include an image of the Irradiated Control

Figure 5a: we include in the graph the Irradiated Control

Figure 6c: we include an image of the Irradiated Control

Figure 10: we included in the main text in line 620: “Irradiated and non-irradiated Control eggs incubated with PBS did not produce any visible change in the vasculature, like Muc control”.

Point 14. Figure 4, the line captions in the graphical figures a and b are not given. What blue and black lines are represented here?

Answer: Thank you for the comment, we modified Figure 4.

Point 15. In each figure authors used ANOVA analysis, however, they have not mentioned any Post-Hoc test. What post-hoc test was applied for comparing the groups?

Answer: Thank you for the comment, we included in the label of each figure the post-hoc test that we applied.

Reviewer 4 Report

The paper contains standard methods and standard approaches to the problem which the authors planned to solve; however it seems it is really reliably prepared,

I have only a few minor questions:

1) Fig 1 why there is a gap and overlap in spectral lines

2) why 5%FBS was used for experiments?

3) how IC50 values were calculated

Author Response

The paper contains standard methods and standard approaches to the problem which the authors planned to solve; however it seems it is really reliably prepared,

I have only a few minor questions:

Point 1. Fig 1 why there is a gap and overlap in spectral lines

Answer: We performed fluorescence emission spectra between 470 – 600 nm to avoid overlap of the excitation peak of curcumin.

Point 2. why 5%FBS was used for experiments?

Answer: FBS 5% is a conventional concentration that we used for nanoparticle incubation with cells to avoid excessive formation of protein corona that could generates changes in nanovesicles cellular recognition and uptake.

Point 3. how IC50 values were calculated

Answer: We included in Section 2.7, more detailed information on the analysis we performed to calculate the IC50 values using the GraphPad software.

Round 2

Reviewer 2 Report

I agree to publish the revised manuscript.

Author Response

Thank you for your useful guidelines.

Reviewer 3 Report

I am convinced by the answers of the Authors. The manuscript has been significantly improved. I would recommend his work for publication after following minor revisions.

The mechanism of PDT must be elaborated in terms of how exactly ROS generated upon light irradiation on PS, here authors must follow the following literature for the mechanism of PDT:

https://www.tandfonline.com/doi/full/10.1080/1040841X.2018.1467876

https://pubs.acs.org/doi/abs/10.1021/acs.biomac.0c00695

https://www.mdpi.com/1420-3049/25/22/5239/htm

Author Response

Thank you for your guidelines. We have included the references you recommended.